# Analysis of Health-Related Quality of Life Reporting in Phase III RCTs of Advanced Genitourinary Tumors

**DOI:** 10.3390/cancers15235703

**Published:** 2023-12-04

**Authors:** Fabrizio Di Costanzo, Fabiana Napolitano, Fabio Salomone, Anna Rita Amato, Gennaro Alberico, Fortuna Migliaccio, Giovanna Pecoraro, Annachiara Marra, Felice Crocetto, Antonio Ruffo, Sarah Scagliarini, Sabrina Rossetti, Livio Puglia, Marilena Di Napoli, Roberto Bianco, Alberto Servetto, Luigi Formisano

**Affiliations:** 1Department of Clinical Medicine and Surgery, University of Naples “Federico II”, 80138 Naples, Italy; fabri.dicostanzo@studenti.unina.it (F.D.C.); fabiana.napolitano@unina.it (F.N.); fabio.salomone@virgilio.it (F.S.); annarita.amato@unina.it (A.R.A.); gennaro.alberico@unina.it (G.A.); for.migliaccio@studenti.unina.it (F.M.); giovann.pecoraro@studenti.unina.it (G.P.); robianco@unina.it (R.B.); alberto.servetto@unina.it (A.S.); 2Department of Neurosciences, Reproductive Sciences and Odontostomatology, University of Naples “Federico II”, 80138 Naples, Italy; annachiara.marra@unina.it (A.M.); felice.crocetto@unina.it (F.C.); 3Dipartimento di Medicina e di Scienze della Salute “Vincenzo Tiberio”, Università degli Studi del Molise, 86039 Termoli, Italy; antonio.ruffo@unimol.it; 4Division of Oncology, Azienda Ospedaliera di Rilievo Nazionale A. Cardarelli, 80131 Naples, Italy; sarah.scagliarini@aocardarelli.it (S.S.); livio.puglia@aocardarelli.it (L.P.); 5Department of Urology & Gynecology, Istituto Nazionale Tumori IRCCS Fondazione G. Pascale, 80131 Naples, Italy; s.rossetti@istitutotumori.na.it (S.R.); m.dinapoli@istitutotumori.na.it (M.D.N.)

**Keywords:** genitourinary cancers, health-related quality of life, patient-reported outcomes, CONSORT-PRO, randomized controlled trials

## Abstract

**Simple Summary:**

In this manuscript, we investigate the reporting rate of health-related quality of life (HRQoL) in phase III RCTs evaluating therapies in advanced genitourinary cancers published between 2010 and 2022. Results showed underreporting of HRQoL and the need to dedicate more attention to its assessment.

**Abstract:**

Background: As recommended in the European Society for Medical Oncology (ESMO) guidelines, assessment of health-related quality of life (HRQoL) should be a relevant endpoint in randomized controlled trials (RCTs) testing new anticancer therapies. However, previous publications by our group and others revealed a frequent underestimation and underreporting of HRQoL results in publication of RCTs in oncology. Herein, we systematically reviewed HRQoL reporting in RCTs testing new treatments in advanced prostate, kidney and urothelial cancers and published between 2010 and 2022. Methods: We searched PubMed RCTs testing novel therapies in genitourinary (GU) cancers and published in fifteen selected journals (Annals of Oncology, BMC Cancer, British Journal of Cancer, Cancer Discovery, Clinical Cancer Research, Clinical Genitourinary cancer, European Journal of Cancer, European Urology, European Urology Oncology, JAMA, JAMA Oncology, Journal of clinical Oncology, Lancet, Lancet Oncology and The New England Journal of Medicine). We excluded trials investigating exclusively best supportive care or behavioral intervention, as well as subgroup or post hoc analyses of previously published trials. For each RCT, we investigated whether HRQoL assessment was performed by protocol and if results were reported in the primary manuscript or in a secondary publication. Results: We found 85 eligible trials published between 2010 and 2022. Only 1/85 RCTs (1.2%) included HRQoL among primary endpoints. Of note, 25/85 (29.4%) RCTs did not include HRQoL among study endpoints. HRQoL results were non-disclosed in 56/85 (65.9%) primary publications. Only 18/85 (21.2%) publications fulfilled at least one item of the CONSORT-PRO checklist. Furthermore, 14/46 (30.4%) RCTs in prostate cancer, 12/25 (48%) in kidney cancer and 3/14 (21.4%) in urothelial cancer reported HRQoL data in primary publications. Next, HRQoL data were disclosed in primary manuscripts of 12/32 (37.5%), 5/13 (38.5%), 5/16 (31.3%) and 5/15 (33.3%) trials evaluating target therapies, chemotherapy, immunotherapy and new hormonal agents, respectively. Next, we found that HRQoL data were reported in 16/42 (38%) and in 13/43 (30.2%) positive and negative trials, respectively. Finally, the rate of RCTs reporting HRQoL results in primary or secondary publications was 55.3% (n = 47/85). Conclusions: Our analysis revealed a relevant underreporting of HRQoL in RCTs in advanced GU cancers. These results highlight the need to dedicate more attention to HRQoL in RCTs to fully assess the value of new anticancer treatments.

## 1. Introduction

Taken together, urothelial, prostate and kidney cancers account for 13% of new cancer diagnoses every year [1]. In the last decades, substantial changes have been introduced in the therapeutic landscape of these tumors following the approval of immunotherapy and novel target therapies [2,3,4,5,6], which has led to substantial improvements in survival outcomes [7,8].

Along with efficacy endpoints, in recent years, the oncology community has dedicated more attention to the assessment of health-related quality of life (HRQoL) domains in phase III randomized controlled trials (RCTs) testing novel treatments in solid malignancies. Indeed, patient-reported outcomes (PROs) in RCTs can offer a comprehensive evaluation of new anticancer treatments. 

This is particularly relevant for patients with urinary tract cancers, for which signs and symptoms related to the cancer diagnosis, such as pain, polyuria, and hematuria, affect multiple life domains, ultimately leading to social issues, psychological distress, irritation and depression [9].

To uphold relevance of HRQoL evaluation in patients receiving anticancer treatments, the European Society for Medical Oncology (ESMO) developed clinical practice guidelines to help oncologists in the assessment of PROs [10]. Furthermore, in 2015, the updated ESMO Magnitude of Clinical Benefit Scale (ESMO-MCBS), which evaluates the benefits of new therapeutic interventions, included HRQoL assessment (based on social, emotional, physical and cognitive function) among primary outcomes [11]. 

Despite an increased awareness of the relevance of HRQoL assessment, when analyzing results of RCTs in genitourinary (GU) cancers, as well as in other solid tumors, the interest is primarily focused on safety data and efficacy endpoints, such as overall survival (OS) and progression-free survival (PFS). 

Previous publications revealed that HRQoL data, generally listed among secondary or exploratory endpoints, are often unpublished, reported in secondary manuscripts or discussed exclusively at international conferences [12,13]. As a result, novel experimental treatments lack an appropriate and extensive evaluation at the patient level, even after drug approval for clinical practice. 

In this study, we investigated the HRQoL reporting in manuscripts of RCTs testing novel treatments in advanced GU cancers, published from January 2010 to December 2022. We explored whether HRQoL assessment was included among primary, secondary or exploratory endpoints and whether HRQoL data were reported in primary or secondary publications. Furthermore, we evaluated the authors’ adherence to the Consolidated Standards of Reporting Trials (CONSORT) checklist extension for PROs collection and reporting in trials, for articles published after February 2013.

## 2. Materials and Methods

### 2.1. Data Collection

We searched phase III RCTs testing novel systemic treatments in advanced urothelial, prostate and kidney cancers published between January 2010 and December 2022 in 15 selected journals: Annals of Oncology, BMC Cancer, British Journal of Cancer, Cancer Discovery, Clinical Cancer Research, Clinical Genitourinary cancer, European Journal of Cancer, European Urology, European Urology Oncology, JAMA, JAMA Oncology, Journal of clinical Oncology, Lancet, Lancet Oncology and The New England Journal of Medicine. The used terms on PubMed were “urothelial cancer” OR “bladder cancer” OR “prostate adenocarcinoma” OR “prostate cancer” OR “kidney carcinoma” OR “renal clear cell carcinoma”; we also selected the additional PubMed filters “Clinical Trial” and, subsequently, “English Language.” The list of selected trials and characteristics of the PubMed search are available in Appendix A. We looked at the abstracts and titles for the first screening. Exclusion criteria were as follows: (1) not phase III trials; (2) trials investigating the impact of supportive therapies, screening methodologies or behavioral interventions; (3) trials not in a metastatic setting; (4) publications of trial design or study protocols; (5) publications disclosing subgroup, subset or post-hoc analysis of previously published RCTs; (6) trials including pediatric population; (7) brief reports. We included trials comparing medical with non-medical treatments—like surgery and radiotherapy—if the drug was tested in the experimental arm [14]. 

For each clinical trial, following data were collected: first author; digital object identifier (DOI); name of the study; journal; year of publication; impact factor (IF) of the journal in the year of publication (retrieved from the Journal of Citation Reports); primary endpoint and study design (superiority or non-inferiority); whether the results were considered positive (experimental arm superior to control arm in superiority trials or non-inferior to control arm in non-inferiority trials) or negative (experimental arm non-superior to control arm in superiority trials or predefined threshold for non-inferiority not met in non-inferiority trials); comparison versus placebo (yes or no); class of drug in the experimental arm; drug used as experimental treatment; site of primary tumor; sources of funding; countries involved in the trial. After establishing the list of included trials, we expanded our search, and we also explored the study protocol, when available as Appendix A, and gathered the information available on the website https://www.clinicaltrials.gov (accessed on 23 February 2023). 

In addition, we also investigated whether the HRQoL endpoint was included in the manuscript or only in the study protocol. When HRQoL results were not reported in primary articles, we searched for secondary publications reporting HRQoL data. Secondary publications were searched in PubMed using the name of the drug and the study’s acronym. The research of secondary publications was not limited to the limited list of journals mentioned above. We collected data about first author, DOI, date of publication, journal, time between primary and secondary publication and IF of the journal in the year of publication. Every trial was scheduled on an electronic database, and each record was reviewed by a different investigator.

Furthermore, for articles published after February 2013, we explored whether they respected the items among those listed in good reporting practice for the Patient-Reported Outcome (PRO) Checklist [15]—available at http://www.consort-statement.org/extensions (accessed on 23 February 2023).

### 2.2. Statistical Analysis

When relevant, statistically significant differences between the groups under analysis were calculated using Fisher’s exact test with GraphPad Prism 9 software. Moreover, Kaplan-Meier curves indicating the probability of HRQoL data publication were obtained using the same program.

## 3. Results

### 3.1. Study Characteristics

Eighty-five phase III RCTs were identified from the PubMed search, evaluating novel therapies in advanced urothelial, prostate and kidney cancers and published between January 2010 and December 2022, that respected searching criteria (Figure 1). The study characteristics are summarized in Table 1. The highest rate of manuscripts was published in The New England Journal of Medicine (29.4%). Target therapies, chemotherapy, hormonotherapy and immunotherapy were investigated in 32 (37.6%), 13 (15.3%), 20 (23.5%) and 16 (18.8%) trials, respectively. Only 12/85 trials (14.1%) were funded by a public institution, while most RCTs (n = 73/85, 85.9%) were funded by pharmaceutical companies. Among the 85 RCTs, 46 (54.1%) tested novel therapies in prostate cancer, 25 (29.4%) focused on kidney cancer and 14 (16.5%) focused on urothelial cancer. Results of the trials were positive in 42/85 (49.4%) cases. We found that 28/85 (32.9%) had blinded masking.

### 3.2. Primary Endpoints and Assessment of HRQoL

OS and PFS were the primary study endpoints in 53/85 (62.4%) and 37/85 (43.5%) RCTs, respectively. Overall response rate (ORR) and safety were assessed among primary endpoints in 1/85 (1.2%) and 2/85 (2.4%) trials, respectively.

Next, we investigated the inclusion of HRQoL among the study endpoints. Of note, only 1/85 trials (1.2%) included HRQoL assessment as a primary endpoint (Table 2). The only RCT evaluating HRQoL as a primary endpoint investigated androgen-deprivation therapy in prostate cancer [16]. 

Remarkably, 25/85 trials (29.5%) did not include any HRQoL assessment among their endpoints. Of these 25 RCTs, 13 evaluated treatments in prostate cancer, eight in kidney cancer and four in urothelial cancer. In addition, HRQoL evaluation was not included in a consistent portion of positive trials (n = 7/42, 16.7%).

Assessment of HRQoL was declared among secondary or exploratory endpoints in 42/85 (49.4%) and 17/85 (20.0) trials, respectively. Health-related quality of life was a secondary endpoint in 58.3% of RCTs involving immunotherapy treatments, in 53.8% of trials regarding target therapies, in 53.8% of RCTs involving chemotherapy and in 60% of RCTs investigating therapies with new hormonal agents. 

Data regarding assessment of HRQoL among study endpoints, based on study characteristics, are summarized in Table 2.

### 3.3. HRQoL Results in Primary Publications

We investigated whether HRQoL results were reported in primary publications. We found that only 34.1% RCTs reported HRQoL results in primary publications (Table 3). HRQoL results were reported in primary publications of 14/46 (30.4%) trials in prostate cancer, of 12/25 (48%) RCTs in kidney cancer and of 3/14 (21.4%) studies in urothelial cancer (Table 3). 

Further analysis showed that RCTs evaluating chemotherapy, target therapies, new hormonal agents and immunotherapy as single agents or combined did not disclose HRQoL data in the primary manuscript in 61.5%, 57.7%, 66.6% and 68.8% of cases, respectively (Table 3). In addition, non-inferiority RCTs disclosed HRQoL data in primary manuscripts in 71.4% of cases, compared to 30.8% of RCTs with a superiority design.

Notably, only 16 out of 42 positive trials (38%) reported HRQoL results in primary manuscripts. In addition, the rate of negative trials disclosing HRQoL data in primary publications was 30.2%.

The characteristics of the studies reporting HRQoL data in primary articles are summarized in Table 3.

### 3.4. HRQoL in Secondary Publications 

For the 85 trials gathered in our analysis, the aggregated probability of HRQoL results publication within 12, 24 and 48 months was 37.9%, 43.7% and 47.1%, respectively (Figure 2A). 

Moreover, we observed a significant disparity in reporting of HRQoL results across positive and negative trials. For positive RCTs, the probability of HRQoL results being published within 12, 24 and 48 months was 52.4%, 61.9% and 72.1%, respectively (Figure 2B). On the other hand, for negative trials, the probability of publishing QoL data within 48 months was 29.5% (Figure 2B).

Among the 56 RCTs without HRQoL results in primary publications, only 18 (31.6%) reported HRQoL data in a secondary article or at an international conference. Of note, only 59.3% positive trials not disclosing HRQoL in primary publication reported HRQoL data in a secondary article. Conversely, only 6.7% trials with negative results published a secondary article disclosing HRQoL results (Figure 3, Table 4). 

A thorough summary of HRQoL assessment disclosing is displayed in Table 5. Of the 85 selected RCTs, 24 disclosed HRQoL assessment in the protocol, not in the manuscript, and 15 (62.5%) reported it in a secondary publication. Fourteen out of 85 RCTs disclosed the HRQoL assessment in the methods section of the primary manuscript, not reporting it in the results section. Of these, seven (50%) reported HRQoL assessment in a secondary publication.

### 3.5. Assessment of CONSORT PRO Items in Publications 

We investigated the compliance with CONSORT PRO items [17] in manuscripts published from 2013, after CONSORT PRO extension was developed. We selected 27 RCTs in which assessment of HRQoL was reported in the primary manuscript. 

Among the items, only 4 out of 27 trials (14.8%) reported statistical approaches for dealing with missing data, while statement of PRO hypothesis with identification of relevant domains was the most present item (13 trials out of 27, 48.1%) (Table 6).

### 3.6. Analysis of HRQoL Reporting per Genitourinary Tumor

Differences in HRQoL reporting emerged among RCTs in prostate, urothelial and kidney cancers. Particularly, of 25 RCTs assessing therapies in kidney cancer, 15 (60%) reported HRQoL assessment among trial endpoints (Table 2), and only 12 (48%) disclosed it in primary publication (Table 3). Among the 13 trials not reporting HRQoL data in primary publications, only four (30.7%) reported data in a secondary publication with an average delay of more than 2 years (774.8 days) (Table 4). 

Among the 14 trials in urothelial cancers, seven (50%) included HRQoL assessment among trial endpoints (Table 2), but only three (21.4%) disclosed them in primary publications (Table 3). Of the 11 RCTs not reporting HRQoL in primary manuscripts, only five (45.5%) disclosed the data in a secondary publication, with an average delay of 821.4 days. 

In advanced prostate cancer, 31 out of 46 trials (67.4%) included HRQoL assessment among endpoints and only 14 (30.4%) reported HRQoL data in primary manuscripts. Of the 32 articles not reporting HRQoL results in primary articles, nine (28.1%) reported them in a secondary publication with an average delay of 918 days.

## 4. Discussion

Only a minor portion of the phase III trials gathered in our research reported HRQoL results in primary publications. Indeed, the percentage of primary papers reporting HRQoL data was lower than the percentage of RCTs not reporting HRQoL data in the primary manuscript (34.1% vs. 65.9%, respectively) (Table 3). Furthermore, when the results were disclosed in a secondary publication, a significant time span between primary and secondary articles was observed.

Taken together, our analysis showed remarkable underreporting of HRQoL data in all three included GU cancers. 

Notably, a consistent fraction of positive RCTs, the trials designed to obtain the approval of novel treatments by regulatory agencies, did not declare HRQoL results in primary or in secondary publications. Therefore, while in most cases, for new anticancer treatments, clinicians have access to efficacy and safety information, a comprehensive assessment is hindered by unavailability of HRQoL data. 

To the best of our knowledge, this is the first study evaluating HRQoL reporting rates in mixed advanced GU cancers. A previous work evaluating publications between 2012 and 2018 found that out of 49 primary publications of trials in prostate cancer declaring HRQoL among study endpoints, HRQoL results were available in 34 (69.4%) and absent in the remaining 15 (30.6%) [12]. However, compared to our analysis, this previous study also included RCTs enrolling patients with non-metastatic/locally advanced prostate cancer, as well as trials evaluating supportive therapies. 

Interestingly, our study reveals that reporting of HRQoL data in metastatic GU cancers is generally lower than it is for other solid cancers. Indeed, the analysis of 446 trials, published between 2012 and 2016, testing novel anticancer treatments for any solid malignancies revealed that HRQoL results were reported in 62.4% of the trials in a metastatic/advanced setting [13]. These rates are higher than those observed herein for kidney, prostate and urothelial cancers (48%, 30.4% and 21.4%, respectively, Table 3). 

Analysis of mature HRQoL results, as well as OS endpoints, generally requires longer time than ORR and PFS. Hence, many RCTs report HRQoL outcomes in secondary, focused articles [18,19]. 

Nevertheless, our research revealed that only 16 (61.5%) of the 26 positive trials not reporting HRQoL data in primary publications disclosed them in a secondary article (Table 4). Additionally, we found that the percentages of profit and non-profit trials that did not include HRQoL assessment among study endpoints were 34.2% and 58.3%, respectively (Table 2), showing that, in line with previous studies [12,13,20,21], non-profit RCTs are more likely than profit RCTs to exhibit this tendency. This discrepancy can have many explanations.

Pharmaceutical corporations typically invest more in academic research due to their higher financial resources, ensuring punctual HRQoL assessment. Moreover, private companies frequently have better knowledge of the prerequisites needed to accelerate the regulatory agencies’ approval for novel drugs. Nevertheless, despite the study protocol of many private-funded RCTs declaring that an HRQoL evaluation would be conducted, the results of HRQoL assessments are not disclosed in a significant portion of publications [20]. 

Furthermore, our study showed that with regard to publishing HRQoL outcomes in primary articles, there is only a small difference between profit and non-profit studies (Table 3).

The aim of many non-profit RCTs designed by academic institutions is towards comprehensive optimization of patients’ care, including validation of new treatment schedules or new doses of an already available drug; for these reasons, a non-inferiority trial design is often adopted.

Thus, HRQoL assessment should be required, particularly in these kinds of RCTs. We discovered that out of the trials we included in our analysis, 4/7 (57.1%) and 3/7 (42.9%) non-inferiority trials were, respectively, profit and non-profit trials. Out of the four profit trials, three (75.0%) published HRQoL data in primary papers, with the remaining RCT not providing HRQoL data in a secondary publication. Out of the three non-profit RCTs, two (66.7%) reported HRQoL results in primary manuscripts, with the other study not publishing HRQoL results in a secondary publication. However, we acknowledge that the low number (n = 7) of the gathered non-inferiority trials may have biased this analysis.

Furthermore, our analysis seems not to show an improving trend in HRQoL reporting through the years. Nonetheless, we were probably prevented from noting it by the heterogeneity of the included RCTs: prostate, kidney and urothelial cancer trials are not aligned in terms of last year’s therapeutical advancements, and further consideration should be given to broader, tumor-focused reports in this genre.

Despite HRQoL not being considered a mandatory endpoint, many regulatory agencies like EMA or FDA, or associations like CONSORT, strongly recommended its assessment within their guidelines and consider it as complementary to efficacy endpoints [22,23]. 

In particular, the CONSORT PROs guidelines were developed and released in 2013 to support investigators in improving the design of clinical trials, enhancing the transparency of findings and aiding physicians in reckoning the benefits of experimental therapies. Our analysis, consistent with previous studies [24], reveals an underreporting of CONSORT-PRO items in publications of RCTs in metastatic genitourinary malignancies (Table 5). 

Specifically, our research showed that the provision of statistical methods for handling missing PRO data was the least represented least represented CONSORT PRO item in primary manuscripts of phase III trials in advanced GU malignancies. In fact, only 11.5% of publications disclosing HRQoL data included this item (Table 6). 

To enhance HRQoL evaluation and reporting in clinical trials, we believe that CONSORT PRO items are critical. Thus, their application for evaluation in trials assessing novel therapies should consider risk of biased results, possibly due to trial design itself. As recent reviews have shown [17,25,26,27,28], a possible source of bias in open-label trials is the patient’s knowledge of their assigned treatment, influencing perception of their symptoms or function and leading to emotion-influenced results far from the patient’s real HRQoL state.

Finally, one of the main obstacles is the heterogeneity of methodologies adopted for measurement of HRQoL in cancer trials. Particularly, several questionnaires have been developed to explore various aspects of HRQoL. We believe that standardization of the methodologies used to assess HRQoL in GU cancers may further clarify the impact of new approved treatments [27]. 

We acknowledge some limitations of this study. We restricted our selection to articles published between 2010 and 2022 and from a limited list of journals. Nonetheless, the most important phase III RCTs in oncology—including those related to genitourinary cancers—are published in the journals covered by our analysis, with a relevant influence on the decisions taken by regulatory agencies. Furthermore, we have chosen the years 2010–2022, which include immunotherapy, target treatments and chemotherapy clinical trials that have impacted the way patients with genitourinary tumors are currently managed.

## 5. Conclusions

Our analysis of phase III RCTs testing novel therapies for advanced kidney, prostate and urothelial cancers showed a low rate of HRQoL endpoint assessment and reporting. Nevertheless, we believe that, along with efficacy endpoints, phase III RCTs led by pharmaceutical corporations or academic institutions should acknowledge the value of including and assessing the novel drug’s impact on health-related quality of life, especially in high-risk patient subgroups.

In addition, we believe that future evaluations of the already published trials on genitourinary malignancies could provide more insight into whether HRQoL advantages also translate into survival benefits [29], aiding medical professionals to thoroughly evaluate the efficacy and safety of new therapies as well as to shape the best treatment strategy.

## Figures and Tables

**Figure 1 cancers-15-05703-f001:**
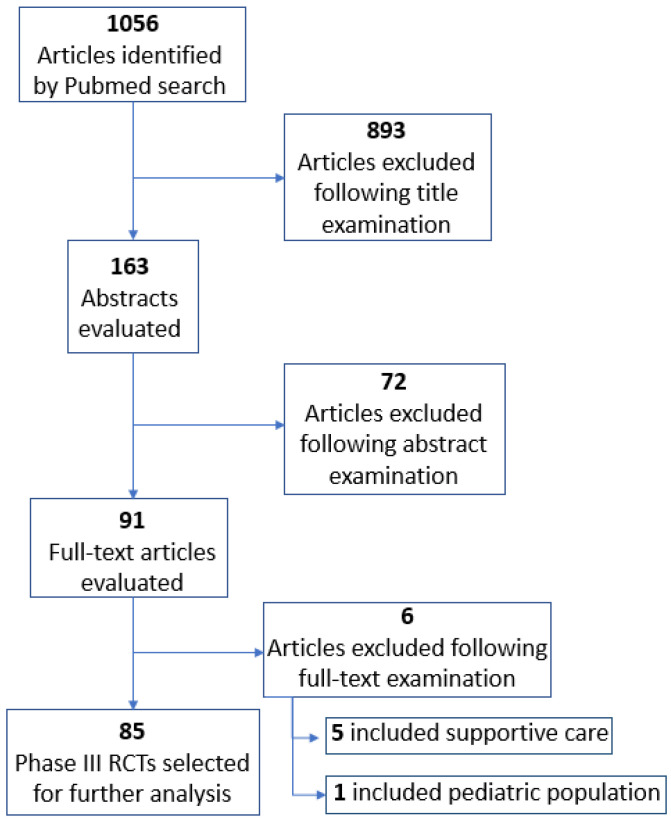
PRISMA diagram for selection of the studies included in the analysis.

**Figure 2 cancers-15-05703-f002:**
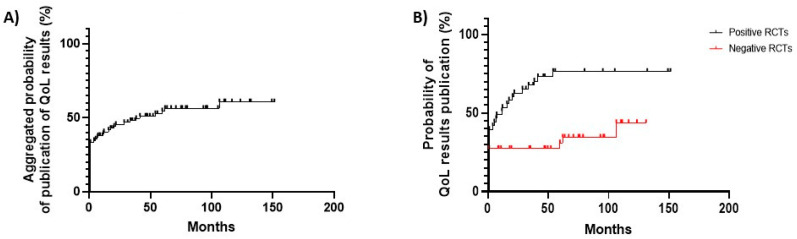
(**A**) Kaplan-Meier curves of time to publication of HRQoL results for all trials gathered in our analysis (n = 85). (**B**) Kaplan-Meier curve of differential time to publication of HRQoL results between positive and negative RCTs (positive trials, n = 42; negative trials, n = 43).

**Figure 3 cancers-15-05703-f003:**
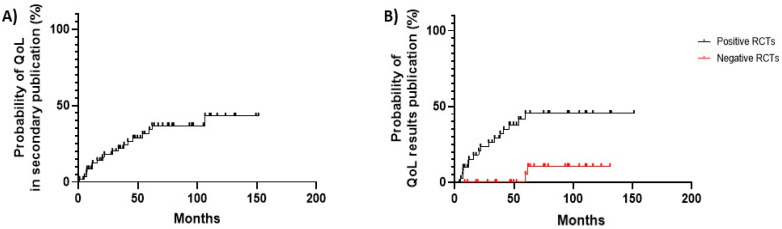
(**A**) Kaplan-Meier curves of time to secondary publication of HRQoL data, for RCTs not reporting HRQoL data in primary publication (n = 57). (**B**) Kaplan-Meier curve of differential time to secondary publication with HRQoL data between positive and negative trials, for RCTs not reporting HRQoL results in primary paper (positive trials, n = 27; negative trials, n = 30).

**Table 1 cancers-15-05703-t001:** Characteristics of the phase III RCTs included in the analysis.

	n	%
Total	85	100
Year of primary publication		
2010	3	3.5
2011	4	4.7
2012	5	5.9
2013	14	16.4
2014	4	4.7
2015	7	8.2
2016	5	5.9
2017	9	10.6
2018	5	5.9
2019	10	11.8
2020	8	9.4
2021	7	8.2
2022	4	4.7
Journal of primary publication		
Annals of Oncology	2	2.4
BMC Cancer	0	-
British Journal of Cancer	0	-
Cancer Discovery	0	-
Clinical Cancer Research	1	1.2
Clinical Genitourinary Cancer	1	1.2
European Journal of Cancer	2	2.4
European Urology	5	5.9
European Urology Oncology	0	-
JAMA	0	-
JAMA Oncology	3	3.5
Journal of Clinical Oncology	21	24.7
Lancet	8	9.4
Lancet Oncology	17	20
New England Journal of Medicine	25	29.4
Class of therapy investigated ^†^		
Immunotherapy	12	14.1
Immunotherapy plus target therapy	4	4.7
Target therapy	26	30.6
Target therapy plus surgery	2	2.4
Chemotherapy	13	15.3
NHA	15	17.6
ADT	5	5.9
Parp-inhibitors	1	1.2
Other *	7	8.2
Control arm: placebo		
Yes	29	34.1
No	56	65.9
Funding		
Profit	73	85.9
Non-profit	12	14.1
Study design		
Superiority	78	91.8
Non-inferiority	7	8.2
Results of the trial		
Positive	42	49.4
Negative	43	50.6
Masking		
Blinded	28	32.9
Open label	57	67.1
Trials included for tumor		
Prostate cancer	46	54.1
Kidney cancer	25	29.4
Urothelial cancer	14	16.5
Primary endpoint ^†,^^		
OS	53	62.4
PFS	37	43.6
ORR	1	1.2
Safety	2	2.4
HRQoL	1	1.2
Other ^‡^	5	5.9

* “Other” included two trials testing antisense oligonucleotides; two including autologous immunotherapies; one including a steroidal antiandrogen; two including experimental vaccines. ^†^ Categories are not mutually exclusive. ^^^ In 15 trials, co-primary endpoints were OS and PFS; in one they were OS and safety; ^‡^ one trial’s endpoint was to assess patient preference between two different treatment schedules; two were other efficacy endpoints; one assessed other AEs; one trial’s endpoint was to test the impact on pain palliation.

**Table 2 cancers-15-05703-t002:** Inclusion of HRQoL among endpoints, based on study characteristics.

	Number of Articles	HRQoL Primary Endpoint n (%)	HRQoL Secondary Endpoint n (%)	HRQoL Exploratory Endpoint n (%)	HRQoL Not Evaluated as Endpoint n (%)
Whole Series	85	1 (1.2)	42 (49.4)	17 (20.0)	25 (29.4)
Year of primary publication					
2010	3	-	1 (33.3)	-	2 (66.7)
2011	4	-	3 (50)	-	1 (25)
2012	5	-	2 (40)	-	3 (60)
2013	14	1 (7.1)	9 (64.3)	2 (14.3)	2 (14.3)
2014	4	-	2 (50)	2 (50)	-
2015	7	-	4 (57.1)	1 (14.3)	2 (28.6)
2016	5	-	2 (40)	-	3 (60)
2017	9	-	5 (55.6)	3 (22.2)	1 (22.2)
2018	5	-	1 (20)	1 (20)	3 (60)
2019	10	-	5 (50)	2 (20)	3 (30)
2020	8	-	5 (62.5)	1 (12.5)	2 (25)
2021	7	-	2 (28.6)	3 (42.8)	2 (28.6)
2022	4	-	1 (25)	2 (50)	1 (25)
Journal of primary publication					
Annals of Oncology	2	-	-	-	2 (100)
BMC Cancer	0	-	-	-	-
British Journal of Cancer	0	-	-	-	-
Cancer Discovery	0	-	-	-	-
Clinical cancer research	1	-	-	-	1 (100)
Clinical Genitourinary Cancer	1	-	-	-	1 (100)
European Journal of Cancer	2	-	2 (100)	-	-
European Urology	5	-	1 (20)	1 (20)	3 (60)
European Urology Oncology	0	-	-	-	-
JAMA	0	-	-	-	-
JAMA Oncology	3	-	-	2 (66.6)	1 (33.3)
Journal of Clinical Oncology	21	-	12 (57.1)	1 (4.8)	8 (38.1)
Lancet	8	-	5 (62.5)	1 (12.5)	2 (25)
Lancet Oncology	17	-	9 (53)	3 (17.6)	5 (29.4)
New England Journal of Medicine	25	1 (4)	13 (52)	9 (36)	2 (8)
Class of therapy investigated ^†^					
Immunotherapy	12	-	7 (58.3)	4 (33.3)	1 (8.3)
Immunotherapy plus target therapy	4	-	1 (25)	2 (50)	1 (25)
Target Therapy	26	-	14 (53.8)	3 (11.5)	9 (34.6)
Target therapy plus surgery	2	-	-	-	2 (100)
Chemotherapy	13	-	7 (53.8)	1 (7.7)	5 (38.5)
NHA	15	-	9 (60)	5 (33.3)	1 (6.7)
ADT	5	1 (20)	3 (60)	-	1 (20)
Parp-inhibitors	1	-	1 (100)	-	-
Other *	7	-	-	2 (28.6)	5 (71.4)
Control arm: placebo					
Yes	29	-	13 (44.8)	7 (24.2)	9 (31)
No	56	1 (1.8)	29 (51.8)	10 (17.8)	16 (28.6)
Primary endpoint ^†,^^					
OS	53	-	22 (41.5)	9 (17)	22 (41.5)
PFS	37	-	16 (43.2)	8 (21.6)	13 (35.1)
ORR	1	-	-	1 (100)	-
Safety	2	-	-	1 (50)	1 (50)
HRQoL	1	1 (100)	-	-	-
Other ^‡^	5	-	4 (80)	1 (20)	-
Funding					
Profit	73	-	36 (49.3)	16 (21.9)	21 (28.8)
Non-profit	12	1 (8.3)	6 (50)	1 (8.3)	4 (33.3)
Study design					
Superiority	78	-	36 (46.1)	17 (21.8)	25 (32)
Non-inferiority	7	1 (14.3)	6 (85.7)	-	-
Results of the trial					
Positive	42	-	24 (57.1)	11 (26.2)	7 (16.7)
Negative	43	1 (2.3)	18 (41.9)	6 (14)	18 (41.8)
Masking					
Blinded	28	-	13 (46.4)	8 (28.6)	7 (25)
Open label	57	1 (1.7)	29 (50.9)	9 (15.8)	18 (31.6)
Trials included for tumor					
Prostate cancer	46	1 (2.2)	23 (50)	9 (19.6)	13 (28.3)
Kidney cancer	25	-	11 (44)	6 (24)	8 (32)
Urothelial cancer	14	-	8 (57.1)	2 (14.3)	4 (28.6)

* “Other” included two trials testing antisense oligonucleotides; two including autologous immunotherapies; one including a steroidal antiandrogen; two including experimental vaccines. ^†^ Categories are not mutually exclusive. ^^^ In 15 trials, co-primary endpoints were OS and PFS; in one they were OS and safety; ^‡^ one trial’s endpoint was to assess patient preference between two different treatment schedules; two were other efficacy endpoints; one assessed other AEs; one trial’s endpoint was to test the impact on pain palliation.

**Table 3 cancers-15-05703-t003:** Disclosure of HRQoL results in primary publications, based on study characteristics.

	Number of Articles	HRQoL Results Reported in Primary Publication n (%)	HRQoL Results Not Reported in Primary Publication n (%)
Whole Series	85	29 (34.1)	56 (65.9)
Year of primary publication			
2010	3	1 (33.3)	2 (66.7)
2011	4	1 (25)	3 (75)
2012	5	2 (40)	3 (60)
2013	14	9 (64.3)	5 (35.7)
2014	4	3 (75)	1 (25)
2015	7	1 (14.3)	6 (85.7)
2016	5	1 (20)	4 (80)
2017	9	4 (44.5)	5 (55.5)
2018	5	2 (40)	3 (60)
2019	10	1 (10)	9 (90)
2020	8	1 (12.5)	7 (87.5)
2021	7	3 (42.9)	4 (57.1)
2022	4	-	4 (100)
Journal of primary publication			
Annals of Oncology	2	-	2 (100)
BMC Cancer	0	-	-
British Journal of Cancer	0	-	-
Cancer Discovery	0	-	-
Clinical cancer research	1	-	1 (100)
Clinical Genitourinary Cancer	1	-	1 (100)
European Journal of Cancer	2	2 (100)	-
European Urology	5	1 (20)	4 (80)
European Urology Oncology	0	-	-
JAMA	0	-	-
JAMA Oncology	3	1 (33.3)	2 (66.7)
Journal of Clinical Oncology	18	4 (28.6)	14 (71.4)
Lancet	8	3 (37.5)	5 (62.5)
Lancet Oncology	20	9 (45)	11 (55)
New England Journal of Medicine	25	9 (36)	16 (64)
Class of therapy investigated ^†^			
Immunotherapy	12	4 (33.3)	8 (66.7)
Immunotherapy plus target therapy	4	1 (25)	3 (75)
Target therapy	26	11 (42.3)	15 (57.7)
Target therapy plus surgery	2	-	2 (100)
Chemotherapy	13	5 (38.5)	8 (61.5)
NHA	15	5 (33.3)	10 (66.6)
ADT	5	3 (60)	2 (40)
Parp-inhibitors	1	-	1 (100)
Other *	7	-	7 (100)
Primary endpoint ^†,^^			
OS	53	16 (30.2)	37 (69.8)
PFS	37	16 (43.2)	21 (56.8)
Safety	1	1 (100)	-
HRQoL	2	-	2 (100)
ORR	1	1 (100)	-
Other ^‡^	5	1 (20)	4 (80)
Control arm: placebo			
Yes	29	9 (31)	20 (69)
No	56	20 (35.7)	36 (64.3)
Funding			
Profit	73	26 (35.6)	47 (64.4)
Non-profit	12	3 (25)	9 (75)
Study design			
Superiority	78	24 (30.8)	54 (69.2)
Non-inferiority	7	5 (71.4)	2 (28.6)
Results of the trial			
Positive	42	16 (38)	26 (62)
Negative	43	13 (30.2)	30 (69.8)
Masking			
Blinded	28	10 (35.7)	18 (64.3)
Open label	57	19 (33.3)	38 (66.7)
Trials included for tumor			
Prostate cancer	46	14 (30.4)	32 (69.6)
Kidney cancer	25	12 (48)	13 (52)
Urothelial cancer	14	3 (21.4)	11 (78.6)

* “Other” included two trials testing antisense oligonucleotides; two including autologous immunotherapies; one including a steroidal antiandrogen; two including experimental vaccines. ^†^ Categories are not mutually exclusive. ^^^ In 15 trials, co-primary endpoints were OS and PFS; in one they were OS and safety; ^‡^ one trial’s endpoint was to assess patient preference between two different treatment schedules; two were other efficacy endpoints; one assessed other AEs; one trial’s endpoint was to test the impact on pain palliation.

**Table 4 cancers-15-05703-t004:** Rate of secondary publications reporting HRQoL results. We only included articles published between 2010 and December 2022 not reporting HRQoL results in primary publications.

	Number of Articles	HRQoL Results Reported in Secondary Publication n (%)	HRQoL Results Not Reported in Secondary Publication n (%)	Average Time From Primary Publication
Articles from 2010 to 2022 not reporting HRQoL results in primary publication	56	18 (31.6)	38 (68.4)	/
Results of the trial				
Positive	26	16 (61.5)	10 (38.5)	797.4 days
Negative	30	2 (6.7)	28 (93.3)	1824.5 days
Type of tumor				
Prostate cancer	32	9 (28.1)	23 (71.9)	918 days
Kidney cancer	13	4 (30.7)	9 (70.3)	774.8 days
Urothelial cancer	11	5 (45.5)	6 (55.5)	821.4 days

**Table 5 cancers-15-05703-t005:** Disclosure of HRQoL assessment in the study protocol or in the methods section of the manuscript.

	HRQoL Assessment Disclosed in the Protocol, Not in the Manuscript n (%)	HRQoL Assessment Disclosed in the Methods Section, Not Reported in the Results Section of the Manuscript n (%)
Whole Series (Total n = 85)	24 (28.2)	14 (16.5)
HRQoL reported in secondary publications		
Yes	15 (62.5)	7 (50)
No	9 (37.5)	7 (50)

**Table 6 cancers-15-05703-t006:** CONSORT PRO checklist items in clinical trial presenting HRQoL data in primary and/or secondary publications from 2013.

	n	%
RCTs reporting HRQoL in primary, secondary publication or both from 2013	27	100
P1b: the PRO should be identified in the abstract as a primary or secondary outcome	6	23.1
P2b: the PRO hypothesis should be stated, and relevant domains identified, if applicable	14	51.9
P6a: evidence of PRO instrument validity and reliability should be provided or cited	6	22.2
P12a: statistical approaches for dealing with missing data are explicitly stated	4	14.8
P20/21: PRO-specific limitations and implications for generalizability and clinical practice	6	23.1

## Data Availability

The list of included trials, along with the Pubmed research string, is available in the Appendix A.

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
