# Peer review of "Analysis of Health-Related Quality of Life Reporting in Phase III RCTs of Advanced Genitourinary Tumors"

_cancers, 2023, doi:10.3390/cancers15235703_

Round 1
Reviewer 1 Report
Comments and Suggestions for Authors
This manuscript reports the results of a review to assess the degree that PRO data is reported in clinical trials. This is very important given that these endpoints can often be exploratory in cancer clinical trials but the dissemination of evidence is so important in shared medical decision making.
-Would recommend using the HRQOL abbreviation throughout the manuscript and not QOL as this is really specific to HRQOL measure.
-Curious why only select journals were identified for the search. These journals likely encompass the majority of clinical trial publications, but the PRO data could be reported in a follow-up manuscript in a different journal after the primary publication.
-Recommend providing more context on the PRO data being published from open-label and single-arm studies because these trial designs are often criticized from the regulatory agencies as producing biased HRQoL results.
Author Response
Please find enclosed our revised manuscript “Analysis of Health-related Quality of Life reporting in phase III RCTs of Advanced Genitourinary tumors” by Di Costanzo et al. We sincerely appreciate helpful critique and input. We enclose herein a revised and hopefully improved manuscript.
This manuscript reports the results of a review to assess the degree that PRO data is reported in clinical trials. This is very important given that these endpoints can often be exploratory in cancer clinical trials but the dissemination of evidence is so important in shared medical decision making.
- Would recommend using the HRQOL abbreviation throughout the manuscript and not QOL as this is really specific to HRQOL measure.
We thank the reviewer for this suggestion. We used the HRQoL abbreviation throughout the manuscript.
- Curious why only select journals were identified for the search. These journals likely encompass the majority of clinical trial publications, but the PRO data could be reported in a follow-up manuscript in a different journal after the primary publication.
We acknowledge that our research is not systematic, due to a choice of selected journals. Of note, the journals selected in our analysis are the ones publishing manuscripts with a potential impact on clinical practice. In addition, the same methodology of research, based on manuscripts published in a selected list of journals, although non-systematic, has been previously used by our and other research teams (PMID: 30304498; PMID: 31734586; PMID: 35259486).
Finally, we clarified the fact that secondary publications, that could include HRQoL results, were not searched in a limited list of journals but in PubMed using the name of the drug and the study’s acronym. We amended the methods section of the revised manuscript with the following statement: “Secondary publications were searched in PubMed using the the name of the drug and study’s acronym. The research of secondary publication was not limited to the limited list of journals mentioned above”.
- Recommend providing more context on the PRO data being published from open-label and single-arm studies because these trial designs are often criticized from the regulatory agencies as producing biased HRQoL results.
We thank the reviewer for this suggestion. We added the following paragraphs in the discussion section: “In our study we found that out of 57 trials with an open label design, 19 reported HRQoL data in primary publication. As carefully discussed in previous works, patients’ awareness of their assigned treatment in clinical trials may represent a potential bias when they fill the HRQoL questionnaires.”
We agree with the reviewer that evaluation of HRQoL in single-arm trials is profoundly biased by the absence of a control arm. Therefore, the academic community, as well as pharma companies, should avoid conducting this kind of analysis since it does not really add value to cancer field.
Thanks for your time, critique and consideration.
Sincerely,
Luigi Formisano
Reviewer 2 Report
Comments and Suggestions for Authors
The aim of this paper is to review the assessment and reporting of health-related quality of life (HRQOL) in Phase III RCTs of advanced genitourinary tumours published between 2010 and 2022 in 15 selected journals. Organisations like the European Society of Medical Oncology (ESMO) and key regulatory bodies all recognise the importance of collecting patient reported outcome (PRO) data so review papers which evaluate how well this is being done are welcome. The Introduction provides an acceptable rationale for the review and I am not aware of any other reviews on this specific topic.
Overall, I think this is a worthy topic for a review but some of the key findings are difficult to pick out. In the Methods, reference is made to investigating whether the QoL endpoint was included in the paper or just in the protocol (line 114). I found it quite hard to extract this information from the results. It would be helpful to very clearly set this out. I would like to know how many trials collected QoL data, according to the protocol, whether the primary paper noted that QoL data were collected and whether the results were reported in the primary paper, a secondary paper or not at all.
A general issue is that some of the results text replicates data presented in the tables e.g. lines 218-223. It would be better to refer to the tables where possible, with the text summarising the findings rather than repeating the data included in the tables.
The terms health-related quality of life (HRQoL) and quality of life (QoL) seemed to be used interchangeably and are not defined. It is important to be consistent with terminology or to explain what the difference is if both terms are used.
The Discussion starts by noting that most of the trials did not adequately assess and report the QoL data. Assessment and reporting are quite different concepts. If the authors want to cover ‘assessment’ they would need to evaluate the instruments were used to measure QoL in these trials, whether the timepoints were sufficient etc. There is some of this, but it would be very useful to report which instruments were used to assess QoL and whether these were generic, cancer specific or diagnosis specific.
It would be good to comment on whether there was evidence of improvement or not in reporting over time in the Discussion.
Specific issues
Line 25 – RCT = randomised controlled (not clinical) trial
Line 29: It would be more accurate to state 15 journals were searched.
The reason the 15 journals were selected should be provided in section 2.1. I was expecting to see the full PubMed search in the supplementary file but it did not seem to be there.
Tables 4 and 5 are the wrong way round.
Lines 232-244 are a long list of results and should not be included in the Discussion here.
The final paragraph of the paper (line 306) are study limitations and not conclusions.
In the PRISMA diagram, it would be good to add the reasons that the 6 full text articles were excluded.
Comments on the Quality of English Language
I appreciate that English is probably not the first language of any of the authors and in general, the standard of English is not bad. However, there are quite a few corrections required throughout the paper. In addition, I think most readers would appreciate shorter paragraphs.
Author Response
Please find enclosed our revised manuscript “Analysis of Health-related Quality of Life reporting in phase III RCTs of Advanced Genitourinary tumors” by Di Costanzo et al. We sincerely appreciate helpful critique and input. We enclose herein a revised and hopefully improved manuscript. Please see the attachment.
Thanks for your time, critique and consideration.
Sincerely,
Luigi Formisano

Round 2
Reviewer 2 Report
Comments and Suggestions for Authors
I think that the authors have satisfactorily addressed the issues raised in my first review.
Comments on the Quality of English Language
There are a few minor errors remaining but generally the quality of the English language was fine.